# Mechanisms Underlying the Protective Effect of Maternal Zinc (ZnSO_4_ or Zn-Gly) against Heat Stress-Induced Oxidative Stress in Chicken Embryo

**DOI:** 10.3390/antiox11091699

**Published:** 2022-08-30

**Authors:** Yunfeng Zhang, Lingyu Xie, Xiaoqing Ding, Yuanyuan Wang, Yibin Xu, Danlei Li, Shuang Liang, Yongxia Wang, Ling Zhang, Aikun Fu, Xiuan Zhan

**Affiliations:** 1Hainan Institute, Zhejiang University, Yongyou Industry Park, Yazhou Bay Sci-Tech City, Sanya 572000, China; 2Key Laboratory of Animal Nutrition and Feed Science in East China, Ministry of Agriculture, College of Animal Sciences, Zhejiang University, Hangzhou 310058, China; 3Key Laboratory of Applied Technology on Green-Eco-Healthy Animal Husbandry of Zhejiang Province, College of Animal Science and Technology, College of Veterinary Medicine, Zhejiang A & F University, Linan 311300, China; 4College of Animal Sciences, Anhui Agricultural University, Hefei 230036, China

**Keywords:** maternal zinc, chick embryo, antioxidant, oxidative stress, Nrf2

## Abstract

Environmental factors such as high temperature can cause oxidative stress and negatively affect the physiological status and meat quality of broiler chickens. The study was conducted to evaluate the effects of dietary maternal Zn-Gly or ZnSO_4_ supplementation on embryo mortality, hepatocellular mitochondrial morphology, liver antioxidant capacity and the expression of related genes involved in liver oxidative mechanisms in heat-stressed broilers. A total of 300 36-week-old Lingnan Yellow broiler breeders were randomly divided into three treatments: (1) control (basal diet, 24 mg zinc/kg); (2) inorganic ZnSO_4_ group (basal diet +80 mg ZnSO_4_/kg); (3) organic Zn-Gly group (basal diet +80 mg Zn-Gly/kg). The results show that maternal zinc alleviated heat stress-induced chicken embryo hepatocytes’ oxidative stress by decreasing the content of ROS, MDA, PC, 8-OHdG, and levels of HSP70, while enhancing T-SOD, T-AOC, CuZn-SOD, GSH-Px, CTA activities and the content of MT. Maternal zinc alleviated oxidative stress-induced mitochondrial damage in chick embryo hepatocytes by increasing mitochondrial membrane potential and UCP gene expression; and Caspase-3-mediated apoptosis was alleviated by increasing CuZn-SOD and MT gene expression and decreasing Bax gene expression and reducing the activity of caspase 3. Furthermore, maternal zinc treatment significantly increased Nrf2 gene expression. The results above suggest that maternal zinc can activate the Nrf2 signaling pathway in developing chick embryos, enhance its antioxidant function and reduce the apoptosis-effecting enzyme caspase-3 activities, thereby slowing oxidative stress injury and tissue cell apoptosis.

## 1. Introduction

Heat stress is a major problem during the hatching of poultry embryos because it results in high mortality and depressed hatchability [1]. Besides, it could lead to oxidative damage of proteins and lipids via affecting mitochondrial function and elevating the levels of reactive oxygen species (ROS), thus causing changes in levels of markers of oxidative stress, such as glutathione peroxidase (GSH-Px), superoxide dismutase (SOD), malondialdehyde (MDA) [2,3]. As heat stress occurs, heat shock protein 70 (HSP70) began to serve as an index of stress and cell protector, hence, it can be used as one of the molecular biomarkers to assess the cell physiological effects of heat stress [4].

Maternal effect is an epigenetic phenomenon that directly affects nutrient metabolism, physiological characteristics, and the growth of offspring [5]. Zinc, an essential trace element, is involved in many cellular biological processes, such as DNA reproduction, cellular respiration, free radical scavenging and maintenance of cell membrane structure [6]. Zinc can induce the generation of metallothionein (MT); MT not only can be used as the repository of zinc but also participate in a variety of physiological processes, including immune function and oxidative stress [7]. Previous studies have found that maternal zinc is first transferred from the liver of broilers to ovaries and developing oocytes, then to egg yolks, and finally to chicken embryonic tissue [8]. Therefore, zinc deposition in eggs from zinc supplementation is required for the rapid growth and complete development of the embryos. In addition, maternal zinc supplement can protect the normal development of progeny embryos from the influence of maternal heat stress [9]. However, the molecular mechanism of the protective effect of maternal zinc on chicken embryos against heat stress is little known. Inorganic ZnSO_4_, zinc oxide, and zinc chloride are common zinc supplement forms in poultry diet. Nevertheless, organic zinc amino acid chelates or complexes are increasingly being used because of their potential higher absorption and bioavailability [10,11]. Our previous study demonstrated that dietary supplementation of either inorganic or organic zinc could enhance the antioxidant enzyme activities and MT concentration to reduce tissue cell oxidative damage in the liver of chicken embryo, and organic zinc had better effects [5]. This may be associated with nuclear factor erythroid 2-related factor 2 (Nrf2) signaling pathway [12]. In a human retinal pigment epithelial cell line study, zinc was also found to increase the biosynthesis of glutathione to protect against oxidative stress through an Nrf2 dependent pathway [13].

In addition, liver cells have the potential for rapid adaptation to heat stress [14]. Therefore, the liver of chicken embryo may be an ideal tissue model in vitro to explore the mechanism of zinc resistance to heat stress. Zinc has been shown to alleviate the adverse effects of heat stress on broilers, but its protective effects on chicken embryos and the mechanism of protection remains unclear. We hypothesized that maternal inorganic zinc sulfate (ZnSO_4_) or organic zinc Glycine (Zn-Gly) might protect chicken embryos from heat stress by enhancing the antioxidant capacity and reducing apoptosis of chicken embryos through an Nrf2-dependent pathway in hepatocytes. Therefore, in this study, Nrf2 signaling pathway was used as the main line to study the effects of maternal zinc (Zn-Gly or ZnSO_4_) on the chicken embryos hepatocellular redox and apoptosis, and further explore its protective mechanism.

## 2. Materials and Methods

### 2.1. Experimental Design, Birds, and Diets

A total of 300, 36-week-old Lingnan Yellow broiler breeders were randomly divided into 3 treatment groups, namely control group, ZnSO_4_ group and Zn-Gly group, each group containing 5 replicates with 20 birds each. During the initiation of the trial period, all broiler breeders were placed in cages (Two breeding chickens per cage), and maintained on an 18 h light and 6 h dark cycle, with a temperature of 26 °C to 30 °C. All broiler breeders were fed a corn-soybean meal basal diet with 24 mg Zinc/kg for 4 weeks pre-test to consume excess zinc in the body. In the first week of the experiment, the control group was fed a corn-soybean meal basal diet, while the ZnSO_4_ group and Zn-Gly group were supplemented with 80mg/kg ZnSO_4_ (80 mg zinc/kg from ZnSO_4_) and 80 mg/kg Zn-Gly (80 mg zinc/kg was derived from crystal standard zinc, which was provided by PANCOSMA, Switzerland.) based the control diet, respectively. The experiment lasted for 8 weeks, and all birds were artificially inseminated. To ensure the high fertilization rate of the eggs, fertilization was adopted once every 3–5 days, and the male-female ratio of chickens during artificial insemination is about 1:50. Table 1 shows the corn-soybean meal basal diet formulation and nutrient content. At the last week of the experiment, 80 eggs per replicate were collected and then incubated under standard conditions in a commercial incubator. On day 17 of incubation, 40 eggs per replicate were randomly selected and placed in another incubator and given 39.5 °C heat stress for 6 h.

### 2.2. Sample Collection

After 6 h of heat stress, 10 eggs that developed normally and survived were randomly selected from each of the high-temperature stimulation groups and the normal hatching groups (6 groups in total) for sample collection. The hatching eggs were opened, the chicken embryos dissected, and the liver was isolated and fixed with 2.5% glutaraldehyde or 4% paraformaldehyde, respectively. The remaining liver tissue sample was stored at −80 °C until further analysis.

### 2.3. Assessment of the Oxidation Status

Approximately 0.5 g of liver tissue from each repetition was homogenized with 4.5 mL of hypothermic saline buffer solution and then homogenates were centrifuged at 3000 r/min at 4 °C for 20 min to obtain the supernatants. The liver oxidation stress relative indicators of ROS, MDA, protein carbonyl (PC), 8-hydroxydeoxyguanosine (8-OHdG), total superoxide dismutase (T-SOD), CuZn-SOD, GSH-Px and CTA, as well as total antioxidant capacity (T-AOC) were tested using the corresponding commercial assay kits (Nanjing Jiancheng Bioengineering Research Institute, Nanjing, Jiangsu, China) and the protein content of the samples was determined by Coomassie brilliant blue method. The level of MT and HSP70 in liver was determined using an enzyme-linked immunosorbent assay (ELISA) kit (Shanghai Enzyme Link Biotechnology Co., Ltd., Shanghai, China). All programs are operated in accordance with the manufacturer’s instructions.

### 2.4. Mitochondrial Morphology

The liver tissue samples were rinsed three times within a 0.1 mol L^−1^ phosphate-buffered saline (pH 7.2) after fixation with 2.5% (wt/vol) glutaraldehyde for 24 h at 4 °C, and then fixed in 1% osmium tetroxide in 0.1mol L^−1^ sodium phosphate buffer (pH 7.2) for 1–2 h, and cleaned ultrasonically (40 kHz) in distilled water for 10 min. A stereomicroscope was used to check if it was clean. The samples were dehydrated by using successive treatments in 30%, 50%, 70%, 80%, 90% and 95% ethanol. Finally, the samples were treated with 100% ethanol and pure acetone for 20 min each. The sample block was infiltrated with 1:1 epon–acetone for 60 min and 3:1 epon–acetone for 180 min. It was subsequently immersed in fresh 100% epon twice for 1 h. After 48 h incubation in a 70 °C vacuum oven, it was sliced to obtain slice thicknesses of 70–90 nm, stained with alkaline lead citrate solution and 50% uranyl acetate saturated solution in ethanol for 5–10 min each and using a transmission electron microscope to observe the section.

### 2.5. Mitochondrial Membrane Potential (MMP)

Weigh 200 mg of fresh and clean chicken embryo liver tissue, cut the tissue into small tissue fragments with scissors, add 2 mL of cold physiological saline, and bath on ice for 3 min. Then, centrifuge at 4 °C for 5 min, discard the supernatant, add about 1.6 mL of low-temperature trypsin digestion solution to the precipitated tissue and mix, ice bath for 20 min, then centrifuge at 4 °C for 5 min, discard the add 2 mL of mitochondrial separation reagent about the tissue weight to the precipitate, and perform ice bath homogenization; centrifuge for 5 min at 4 °C, take the supernatant, and carefully transfer the supernatant to another clean centrifuge tube. Centrifuge for 10 min at 4 °C, discard the supernatant, and the pellet is mitochondria. The protein concentration of the isolated mitochondrial samples was determined by the BCA protein concentration assay kit (Shanghai Beyotim Bioengineering Research Institute, Shanghai, China), and all procedures were operated in accordance with the instructions. Then, add 0.1 mL of purified mitochondria with a total protein content of 10–100 μg to 0.9 mL of 5-fold diluted JC-1 staining working solution. After mixing, time scan was performed directly with a fluorescence spectrophotometer, the excitation wavelength was 485 nm, and the emission wavelength was 590 nm. Finally, it was observed with a fluorescence microscope.

Weigh 200 mg of fresh and clean chicken embryo liver tissue, cut the tissue into small tissue fragments with scissors, add 2 mL of cold physiological saline, and bath on ice for 3 min. Then, centrifuge at 4 °C for 5 min, discard the supernatant, add about 1.6 mL of low-temperature trypsin digestion solution to the precipitated tissue and mix, ice bath for 20 min, then centrifuge at 4 °C for 5 min, discard the add 2 mL of mitochondrial separation reagent about the tissue weight to the precipitate, and perform ice bath homogenization; centrifuge for 5 min at 4 °C, take the supernatant, and carefully transfer the supernatant to another clean centrifuge tube. Centrifuge for 10 min at 4 °C, discard the supernatant, and the pellet is mitochondria. The protein concentration of the isolated mitochondrial samples was determined by the BCA protein concentration assay kit (Shanghai Beyotim Bioengineering Research Institute, Shanghai, China), and all procedures were operated in accordance with the instructions. Then, add 0.1 mL of purified mitochondria with a total protein content of 10–100 μg to 0.9 mL of 5-fold diluted JC-1 staining working solution. After mixing, time scan was performed directly with a fluorescence spectrophotometer, the excitation wavelength was 485 nm, and the emission wavelength was 590 nm. Finally, it was observed with a fluorescence microscope.

### 2.6. Assessment of Hepatocyte Apoptosis and Caspase3 Activity

1. Material: Fresh liver tissue is fixed at 4% paraformaldehyde for more than 24 h. The tissue is then removed in a fume hood and the tissue is trimmed with a scalpel, and the repaired tissue and the corresponding label are placed in the dehydration box. 2. Dehydration: The dehydration box is dehydrated in gradient alcohol in turn. (The dehydration order is 75% alcohol 4 h, 85% alcohol 2 h, 90% alcohol 2 h, 95% alcohol 1 h, absolute ethanol I 30 min, absolute ethanol II 30min, alcohol benzene 5–10 min, xylene I 5–10 min, xylene II 5–10 min, wax I 1h, wax II 1h and wax III 1h). 3. Embedding: First put the melted wax into the embedding frame, and before the wax solidifies, the tissue is removed from the dehydration box and placed in the embedding frame and labeled accordingly. Cool in a refrigerator at −20 °C, remove the wax block from the embedding frame and trim the wax block after solidification. 4. Slice: Place the trimmed wax block on the paraffin microtome and slice it, the thickness of the slice is 4 μm. The prepared paraffin sections were detected for hepatocyte apoptosis using deoxynucleotide-terminal transferase-mediated notched end markers (TUNEL). The activity of Caspase3 in liver was determined using an enzyme-linked immunosorbent assay (ELISA) kit (Shanghai Enzyme Link Biotechnology Co., Ltd., Shanghai, China). All programs are operated in accordance with the manufacturer’s instructions.

### 2.7. qRT-PCR Analysis

Total RNA from each sample was extracted by using Trizol^®^ Regent (Thermo fisher scientific Inc., Waltham, MA, USA), and the concentration and purity of the extracted RNA were determined by NanoDrop 2000 Spectrophotometer (Thermo fisher scientific Inc., Waltham, MA, USA). The integrity of the extracted RNA was detected by 1% agarose gel electrophoresis, and cDNA was synthesized using the Prime Script RT Reagent Kit with gDNA Eraser (Takara Biomedical Technology Co., Ltd., Tokyo, Japan) following the manufacturer’s protocols. Gene expression was detected on an ABI 7500 Real-Time PCR System (Bio-Rad, CA, USA) with a Cham QTM Universal SYBR qPCR Master Mix (VA zyme Biotech, Nanjing, China) in accordance with the manufacturer’s protocols. Gallus superoxide dismutase (Gallus SOD), Gallus metallothionein (Gallus MT), Gallus bcl-2-associated x protein (Gallus BAX), Gallus uncoupling protein3 (Gallus UCP3), Gallus nuclear factor erythroid 2-related factor 2 (Gallus Nrf2), and Gallus 18srDNA (Gallus 18S) specific primer sequences (Table 2) were designed according to the national center for biotechnology information (NCBI) database retrieval sequence, the gene expression level was standardized with Gallus 18srDNA (Gallus 18S), and the relative gene expression levels of all target genes were calculated using the 2^−∆∆Ct^ method.

### 2.8. Statistical Analysis

All data were analyzed using SPSS 20.0 (SPSS Inc., Chicago, IL, USA) statistical software and expressed as mean values ± standard error. The main effects of maternal zinc and incubation temperature were tested by Generalized Linear Mode (GLM) model. *p*-value < 0.05 was considered statistically significant. Figures were made by GraphPad Prism 8.00 software (GraphPad Software, San Diego, CA, USA).

## 3. Results

### 3.1. Embryo Mortality

The effects of maternal zinc on chicken embryo mortality under heat stress are shown in Table 3. Compared with the control group and ZnSO_4_ group, Zn-Gly group significantly reduced (*p* < 0.05) the mortality of hatching eggs in the final stage and the whole period. There were no significant interactions between heat stress and maternal zinc treatments (*p* > 0.05).

### 3.2. Liver Oxidation Indicators

The effects of maternal zinc on oxidative indexes and HSP70 levels in embryos under heat stress are shown in Table 4. The oxidation index of ROS, MDA, PC and 8-OHdG and HSP70 contents were markedly elevated (*p* < 0.05) in heat stress-challenged conditions, but the broiler breeders that received maternal zinc treatments decreased (*p* < 0.05) the contents of MDA, PC, 8-OHdG and HSP70 in liver of chicken embryo when compared with the control group. In addition, the Zn-Gly group also decreased (*p* < 0.05) the ROS content compared with both control group and the ZnSO_4_ group and decreased (*p* < 0.05) the contents of ROS, MDA, PC, 8-OHdG and HSP70 in liver of chicken embryo in comparison with ZnSO_4_ group. Meanwhile, there was a significant interaction between heat stress and maternal zinc treatment on the MDA and HSP70 content in liver of chicken embryo (*p* < 0.05).

### 3.3. Liver Antioxidant Indicators

The effects of maternal zinc on anti-oxidative indexes and MT were presented in Table 5. Heat stress significantly decreased the activities of T-SOD, CuZn-SOD, GSH-Px, CAT, T-AOC and the content of MT (*p* < 0.05) in chicken embryo liver, while maternal zinc treatment increased (*p* < 0.05) the above indicators, and Zn-Gly group had a better effect than that of ZnSO_4_ group (*p* < 0.05). Meanwhile, there was an interaction between heat stress and maternal zinc treatment of liver MT content (*p* < 0.05).

### 3.4. Hepatocellular Mitochondrial Morphology

As shown in Figure 1, the mitochondria of chick embryo hepatocytes swelled and some mitochondrial cristae disappeared in the control group under non-heat status, but the mitochondrial morphology was normal in the ZnSO_4_ group and the Zn-Gly group. Under heat stress conditions, mitochondrial vacuolar degeneration, and outer membrane rupture in the control group; mitochondria were swollen, mitochondrial cristae were blurred, and some vacuolar degeneration appeared in the ZnSO_4_ group; mitochondria in the Zn-Gly group were swollen, but the structure was intact. The results suggest that maternal zinc treatment decreased the damaging effect of heat stress on hepatocyte mitochondria to a certain extent, and Zn-Gly group had better effect than that of the ZnSO_4_ group (*p* < 0.05).

### 3.5. Hepatocellular MMP, Apoptosis index (AI), and Caspase3 Activity

The effects of maternal zinc on MMP and apoptosis of hepatocytes are shown in Table 6 or Figure 2. Heat stress significantly decreased (*p* < 0.05) MMP in hepatocytes and increased (*p* < 0.001) the AI and caspase3 activity of hepatocytes, while the maternal zinc treatment significantly improved the above indexes compared with the control group (*p* < 0.001), and Zn-Gly group had better effect than that of the ZnSO_4_ group (*p* < 0.05). Meanwhile, there was an interaction between heat stress and maternal zinc treatment on the AI of hepatocytes (*p* < 0.001). Regardless of heat stress or non-heat stress, the AI of chick embryo hepatocytes in the Zn-Gly group were the lowest.

### 3.6. Expressions of Embryonic Hepatocyte Gene

The effect of maternal zinc on gene expression of oxidative stress chicken embryo hepatocytes is shown in Table 7. Compared with the control group, heat stress decreased (*p* < 0.05) the abundance of CuZn-SOD and MT mRNA expression in chicken embryo hepatocytes and increased (*p* < 0.001) the relative abundance of Nrf2, uncoupling protein (UCP), and Bax, while maternal zinc treatment increased (*p* < 0.001) Nrf2 and UCP mRNA expressions, decreased (*p* < 0.001) Bax mRNA abundance, and Zn-Gly treatment also increased (*p* < 0.05) CuZn-SOD and MT mRNA abundance. Compared with the ZnSO_4_ group, the Zn-Gly group increased (*p* < 0.05) the mRNA abundance of CuZn-SOD, MT, Nrf2, and UCP mRNA, decreased (*p* < 0.05) the abundance of Bax mRNA (*p* < 0.05). Meanwhile, there was an interaction between heat stress and maternal zinc treatment on CuZn-SOD, UCP, and Bax gene expression (*p* < 0.05). Under heat stress conditions, the Zn-Gly group had the highest CuZn-SOD, UCP mRNA abundance, and lowest Bax mRNA abundance.

## 4. Discussion

Heat stress causes some physiological changes in chicken embryos, such as oxidative stress, which leads to increased mortality and reduced hatching rate [15]. The optimum incubation temperature for chicken embryo development was 37.8 °C [16]. When the incubation temperature reached 40.6 °C from 16 days to 18.5 days of the chick embryo stage, the dead embryo rate increased, and the hatching rate decreased [17]. Similar results were illustrated by Sozcu et al. [15], who reported that continuous high temperature (38.9–40.0 °C) treatment of hatching eggs from 10 days to 18 days of the embryo stage significantly reduced hatching rates. However, our experiments found that heat stimulation at 39.5 °C for 6h on the 17th day of hatching did not cause a significant change in the dead embryo rate. It showed that the thermal stimulation time had different effects on the hatching rate of eggs, and chicken embryos had strong adaptability to short-term heat stimulation.

The antioxidative enzyme system (including T-SOD, CuZn-SOD, GSH-Px, CAT, and T-AOC) acts as the first line of defense against antioxidant [18]. Altering the activity of these enzymes can change the balance between ROS production and the antioxidant system [19]. In the present study, heat stress significantly decreased the activity of major antioxidant enzymes in the liver of chicken embryo. This may be related to the excessive temperature leading to the accumulation of ROS, which exceeds the tolerance threshold level of the body, thus damaging the antioxidant system and leading to the reduction in antioxidant enzyme activity [20]. Therefore, this suggests that the oxidative balance of chick embryos has been disturbed by heat stress. In addition, this study also investigated whether excessive ROS induced by heat stress caused oxidative damage via measuring MDA and PC, and 8-OHdG content in chicken embryo liver. The reason is that MDA, PC and 8-OHDG can be used as markers to evaluate the degree of oxidative damage of lipid, protein, and DNA in animals, respectively [21,22,23,24]. In our experiment, heat stress (39.5 °C, 6 h) significantly increased the contents of ROS, MDA, PC, 8-OHdG in liver of chicken embryo. Thus, these observations suggested that exposure of chicken embryos to heat stress results in oxidative damage of DNA, proteins, and lipids in hepatocytes. In addition, rapid synthesis of HSP is another endogenous mechanism for living cells to adapt to heat stress [25]. Consistent with the present study, previous studies reported that increased expression of HSP70 upon oxidative stress [26]. The reason may be that HSP70, as an intracellular protein chaperone, can sense oxidative damage and play a protective role in cells under various stresses [27].

Mitochondria are the energy supply and metabolic center of cells [28]. The mitochondrial respiratory chain establishes MMPs on both sides of the inner mitochondrial membrane by electron transfer and generates a small amount of ROS [29]. However, in the case of redox imbalance, the excess ROS produced cannot be cleared in time, eventually leading to MMP decline and mitochondrial dysfunction [30]. At the same time, ROS generated by oxidative stress can activate the activity of uncoupling protein (UCP) [31]. UCP is a mitochondrial anion transporter protein that maintains mitochondrial redox balance and reduces the production of ROS [32]. Moreover, excessive ROS can also induce apoptosis through cell death stimulation, involving activation of Bax protein or Caspases3 [33]. In the present study, heat stress disrupted mitochondrial morphology, decreased MMP, and increased UCP and Bax mRNA abundance expression, as well as Caspase3 enzymatic activity. These may be related to the overproduction of ROS triggered by heat stress. In addition, oxidative stress, as an intracellular signaling molecule, can also regulate cellular responses via activating various signaling pathways, such as the Nrf2 pathway [34]. Nrf2 is a key factor in oxidative stress response of cells and is regulated by Keap1 [35]. When heat stress occurs, Nrf2 can protect cells from oxidative stress by regulating gene expression of antioxidant proteins and phase II detoxification enzymes, such as CuZn-SOD and MT [36,37]. CuZn-SOD and MT are considered as endogenous ROS scavengers that confer protection against oxidative stress in vivo [38,39]. Related studies have shown that heat stress can lead to increased CuZn-SOD and MT mRNA abundance expression in hepatocytes [40]. Against our experimental results, this may be due to the destructive effect of excessive ROS on post-transcriptional Nrf2.

Zinc is an essential trace element for poultry, and dietary zinc deficiency increased embryonic mortality and decreased hatchability and healthy chick ratios [16]. Similar results were reported by Huang et al. [41], who demonstrated that severe zinc deficiency in broiler breeder diets results in a lower hatchability rate, abnormal embryonic development, and poor performing offspring. Meanwhile maternal zinc supplementation can not only eliminate these adverse effects, but also improve egg hatchability [42]. In this study, the addition of Zn-Gly could significantly reduce the rate of dead embryo and increase the hatching rate of fertilized eggs in the late stage and the whole stage of hatching chicken embryo. As we all know, the transfer of trace minerals from the hen to the egg involves two pathways: one through the ovaries to the yolk, and the other through the oviduct to the white, shell membrane and eggshell [8]. Among them, yolk zinc deposition was increased with dietary zinc supplementation, and the yolk zinc can be absorbed by the yolk sac membrane and transported to the embryonic liver for storage [43,44]. Moreover, as the yolk sac develops and matures in the later stages of incubation, the amount of zinc transferred to the embryonic liver increases significantly [45]. Therefore, zinc glycine reduced the mortality of late embryo, and whole embryo may be related to dietary zinc supplementation and the degree of yolk sac development in late embryo. In addition, we found that the embryo mortality of Zn-Gly group was significantly lower than that of ZnSO_4_ group. This indicates that zinc fed in organic form has better bioavailability than zinc fed in inorganic form. Similarly, Jarosz et al. [46] showed that supplementation of broiler chickens with Zn-Gly resulted in higher zinc concentrations in the liver compared to ZnSO_4_.

Accumulating evidence shows that zinc deficiency increases free radical production and oxidative damage to proteins, DNA, lipids [47,48], increasing the susceptibility of animals to heat stress. Therefore, zinc plays an important role under heat stress conditions. Zinc can effectively attenuate the heat-shock responses with down-regulated expressions of HSP70 mRNA in broilers [40]. Meanwhile, zinc can also adjust abnormal levels of the MDA and 8-OHdG in tissue and serum to decrease the oxidative damage caused by heat stress in broilers [49]. In our experiment, maternal zinc significantly reduced ROS, MDA, PC, 8-OhdG, and HSP70 contents in chicken embryo liver under oxidative stress. The reason may be that zinc plays an antioxidant role by inhibiting the activity of nicotinamide adenine dinucleotide phosphate oxidase [50]. Our experimental study also shows that the effect of Zn-Gly addition is significantly better than that of ZnSO_4_, these findings are similar to those of Sahin et al. [51], who reported that organic zinc is more effective than ZnSO_4_ at alleviating the negative effects of heat stress-induced oxidative stress in quail. The reason may be that the organozinc is absorbed in the intact form, and the zinc atoms in the organozinc can be safely combined in the organic molecular structure during the absorption process, thereby increasing zinc absorption and utilization [52]. Under normal physiological functions, the body is in a state of redox balance. This is because there are antioxidant protection mechanisms in the body, including a series of antioxidant substances (such as MT) and antioxidant enzymes (such as SOD, GSH-Px, CAT) [53], and their expression levels represent the antioxidant ability of animal. Studies have confirmed that dietary zinc supplementation can increase the activity of T-AOC, CuZn-SOD, GSH-Px, CAT, T-SOD, and the level of MT in various tissues of chickens [34,49,54,55,56]. Consistent results were also obtained in this experiment, maternal zinc treatment increased the activity of antioxidant enzymes and the level of MT in chicken embryo liver under oxidative stress and non-oxidative stress conditions. It is well known that zinc can be transferred from the diet to the embryonic liver, so the increase in antioxidant enzyme activity and MT content in chick embryos may be related to dietary zinc supplementation. For example, Bun et al. [57] showed that diets supplemented with 0, 20, 40 and 60 mg/kg methionine hydroxyl analogs zinc chelates, respectively. The activity of CuZn-SOD increased successively. In addition, many previous clinical studies have shown that supplementation of organic zinc significantly increased the activity of GSH-Px, CuZn-SOD, T-SOD, T-AOC and the content of MT in animal tissues compared with inorganic zinc [58,59,60]. Which is similar to our experimental results. This may be attributed to the fact that the bioavailability of organic zinc in broiler breeders is higher than ZnSO_4_ [61]. Consistent with this explanation, Zhang et al. [5] indicated that the reason for this is that Zn-Gly has a high bioavailability, resulting in increased zinc storage in the embryonic liver, thus the effect of Zn-Gly on the antioxidant capacity of chicken embryo liver was better than that of ZnSO_4_.

Zinc can relieve mitochondrial dysfunction induced by high stress [62,63]. In this experiment, maternal zinc improved mitochondrial morphology and increased MMP in chick embryo hepatocytes, and the effect of Zn-Gly was better than that of ZnSO_4_. The relationship in mitochondria between MMP, mitochondrial morphology, and heat stress has been discussed previously. Therefore, the reason may be related to the increase in antioxidant enzyme activity and content of antioxidant substances and the decrease in ROS level by zinc treatment. In addition, this study also involved measurement of the expression level of UCP and Bax gene, as well as Caspase3 activity, to assess whether zinc can effectively alleviate the damage to mitochondria caused by heat stress. The results show that maternal zinc significantly reduced the abundance of UCP and Bax mRNA and the activity of Caspase3 in chicken embryo liver mitochondria. Overall, maternal zinc attenuated oxidative stress-induced mitochondrial damage in chick embryo hepatocytes by increasing MMP and UCP gene expression, and attenuated apoptosis by increasing Bax gene expression and decreasing caspase3 activity. We also found that maternal zinc significantly reduced the AI of chicken embryo hepatocytes, and the effect of Zn-Gly was better than ZnSO_4_, which may be related to the high absorption and utilization rate of Zn-Gly. Zinc can reduce the damage of heat stress to the liver by the Nrf2 antioxidant signal transduction pathway, increasing the gene expression of Nrf2, and enhancing the antioxidant function [64,65]. This study also confirmed the above conclusion. Previous studies showed the induction of Nrf2 up-regulation by zinc along with a protection against oxidative damage in vivo [66]. For instance, zinc treatment significantly upregulated the expression and function of Nrf2 in mouse vivo [64]. McMahon et al. [67] also reported that Nrf2 expression was induced by 100 and 300 μm zinc in mouse embryonic fibroblasts. However, few studies have reported that the effect of maternal zinc on the Nrf2 pathway in chick embryos. In the present study, maternal zinc increased the expression of Nrf2 gene in chicken embryo hepatocytes, this may be because both Nrf2 expression and its transcriptional function require zinc, and the nuclear translocation of Nrf2 is impaired in zinc-deficient cells, thus affecting its expression. The effect of zinc on Nrf2 signal transduction pathway may be related to the increase in protein kinase B activity [68]. Zinc can maintain the activation of protein kinase B to increase the accumulation of Nrf2 in the nucleus and enhance the expression of its downstream antioxidants, thereby exerting its anti-oxidative damage effect [66]. Zinc is an inducer of MT and an important component of CuZn-SOD. We found that maternal zinc decreased the mRNA expression of MT and CuZn-SOD in chicken embryo liver; this echoes the result of increased antioxidant enzyme activity in this study because these mRNAs have the effect of scavenging free radicals [9]. Therefore, these results suggest that maternal zinc can prevent oxidative damage caused by heat stress by affecting endogenous antioxidants and stress proteins.

## 5. Conclusions

This study demonstrates the protective mechanism of maternal zinc against oxidative stress induced by heat stress in chicken embryo. Maternal zinc reduced the mortality of chicken embryos by activating the Nrf2 signaling pathway in developing chicken embryos, enhancing its antioxidant function, and reducing the activity of the apoptosis-affecting enzyme caspase-3, slowing down the oxidative stress damage and apoptosis of tissue cells, and Zn-Gly group had better effect than that of ZnSO_4_. This study provides new insights into the protective mechanism of maternal zinc on the antioxidant defense of poultry embryos.

## Figures and Tables

**Figure 1 antioxidants-11-01699-f001:**
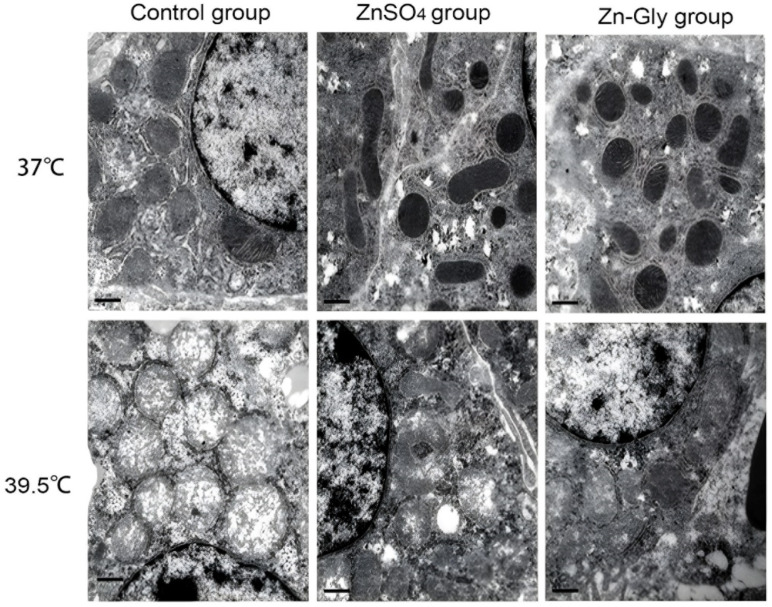
Effect of maternal zinc on hepatocellular mitochondrial morphology in embryos under heat status (30,000×).

**Figure 2 antioxidants-11-01699-f002:**
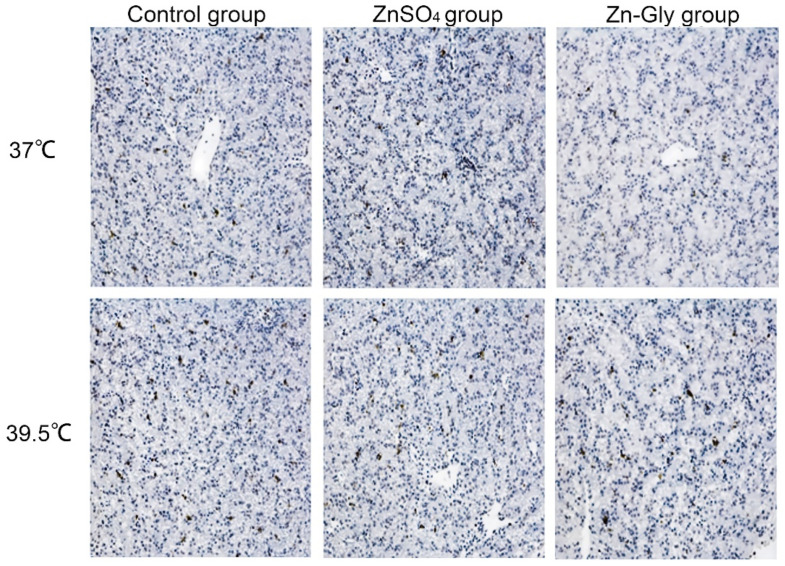
Effect of maternal zinc on liver apoptosis in embryos under heat stress (200×).

**Table 1 antioxidants-11-01699-t001:** Composition and nutrient level of the basal diet (g/kg, unless otherwise stated).

Ingredients (%)	g/kg of Total Diet
Maize	646
Soybean meal	250
CaHPO_4_	18
Limestone	70
Salt	3
DL-methionine	3
Premix ^1^	10
Nutrient Composition	
ME ^2^ (MJ/kg)	11.20
Crude protein	161.1
Calcium	30.4
Total phosphorus	6.3
Lysine	8.0
Met + Cys	8.2
Zine (mg/kg)	24

^1^ Premix provided the following per kilogram of diet: VA, 10,800 IU; VD3, 2160 IU; VE, 27 IU; VK3, 1.4 mg; VB1, 1.8 mg; VB2, 8 mg; VB6, 4.1 mg; VB12, 0.01 mg; niacin, 32 mg; pantothenate, 11 mg; folic acid, 1.08 mg; biotin, 0.18 mg; choline, 450 mg; Fe, 80 mg; Cu, 8 mg; Mn, 100 mg; I, 1.0 mg; Se, 0.3 mg. ^2^ ME based on calculated values; others were analyzed values.

**Table 2 antioxidants-11-01699-t002:** Primers used for the target and reference.

Genes	Genbank Accession	Forward Sequence	Reverse Sequence
SOD	U28407.1	5’-CTGGAAATGCTGGACCTCGTTTAG-3’	3’-CAGCTCATTTCCCACTGCCATCTT-5’
MT	X06749.1	5’-GAGCCGAACCGACCCGAACT-3’	3’-CTTGCACGACCCAGCACAGGA-5’
BAX	FJ977571.1	5’-GGATTCTCACAGTAGGAGGATGGAT-3’	3’-GGCCACCAGTGAAGGCAAAC-5’
UCP3	NM_204107.1	5’-GCAGAGAAACAGAGCGGGATTTGA-3’	3’-GGCTCCTGGCTCACGGATAGA-5’
Nrf2	NM_205117.1	5’-GTGCCGCAGGGCAATGCTAGT-3’	3’-GCCAGCAGGAGGGTCTTTCTTTG-5’
18S	AF173612.1	5’-CCGGACACGGACAGGATTGACA-3’	3’-CAGACAAATCGCTCCACCAACTAAG-5’

SOD, superoxide dismutase. MT, metallothionein. BAX, bcl-2 associated X protein. UCP3, uncoupling protein3. Nrf2, nuclear factor erythroid 2-related factor 2. 18S, 18S rDNA.

**Table 3 antioxidants-11-01699-t003:** Effects of maternal zinc on embryo mortality under heat stress (%) (mean ± S.E.).

Temperature	Maternal Zinc	Late Mortality	Whole-Term Mortality
37 °C ^1^	Control	6.00 ± 0.58 ^a^	8.57 ± 0.51 ^a^
	ZnSO_4_	5.34 ± 0.21 ^a^	7.55 ± 0.54 ^ab^
	Zn-Gly	3.86 ± 0.10 ^b^	6.02 ± 0.89 ^b^
39.5 °C ^1^	Control	6.03 ± 0.89 ^a^	8.60 ± 0.97 ^a^
	ZnSO_4_	4.89 ± 0.80 ^ab^	7.53 ± 1.26 ^ab^
	Zn-Gly	3.85 ± 1.20 ^b^	6.01 ± 0.72 ^b^
37 °C ^2^		5.07 ± 0.99	7.38 ± 1.25
39.5 °C ^2^		4.92 ± 1.27	7.38 ± 1.43
	Control ^3^	6.01 ± 0.68 ^a^	8.59 ± 0.69 ^a^
	ZnSO_4_ ^3^	5.12 ± 0.58 ^a^	7.54 ± 0.87 ^a^
	Zn-Gly ^3^	3.86 ± 0.76 ^b^	6.02 ± 0.72 ^b^
*p*-value			
Temperature		0.687	0.996
Maternal zinc		0.001	0.001
Temperature × maternal zinc		0.833	0.999

Different letters (a–b) represent significant differences between the groups. ^1^ Each value represents the mean ± SE of 5 replicates (*n* = 5). ^2^ Each value represents the mean ± SE of 15 replicates (*n* = 15). ^3^ Each value represents the mean ± SE of 10 replicates (*n* = 10).

**Table 4 antioxidants-11-01699-t004:** Effects of maternal zinc on oxidative indicators and HSP70 in embryos under heat stress (mean ± S.E.).

Temperature	Maternal Zinc	ROS, nmol/mg port	MDA, nmol/mg port	PC, nmol/mg port	8-OHdG, nmol/mg port	HSP70, ng/g
37 °C ^1^	Contro	1396 ± 56.61	0.60 ± 0.05 ^bc^	0.99 ± 0.19 ^bc^	14.14 ± 0.91 ^cd^	14.14 ± 0.91 ^cd^
	ZnSO_4_	1325 ± 61.28	0.58 ± 0.02 ^bc^	0.88 ± 0.13 ^cd^	14.09 ± 0.52 ^cd^	14.09 ± 0.52 ^cd^
	Zn-Gly	1196 ± 52.02	0.53 ± 0.07 ^c^	0.71 ± 0.05 ^d^	13.08 ± 0.45 ^d^	13.08 ± 0.45 ^d^
39.5 °C ^1^	Control	1553 ± 48.64	0.97 ± 0.13 ^a^	1.49 ± 0.14 ^a^	33.08 ± 1.12 ^a^	33.08 ± 1.12 ^a^
	ZnSO_4_	1544 ± 32.14	0.69 ± 0.01 ^b^	1.17 ± 0.00 ^b^	22.66 ± 0.84 ^b^	22.66 ± 0.84 ^b^
	Zn-Gly	1495 ± 74.95	0.56 ± 0.04 ^c^	1.0 ± 0.04 ^bc^	15.16 ± 1.45 ^c^	15.16 ± 1.45 ^c^
37 °C ^2^		1295 ± 102.68 ^b^	0.57 ± 0.05 ^b^	0.86 ± 0.17 ^b^	13.77 ± 0.78 ^b^	13.77 ± 0.78 ^b^
39.5 °C ^2^		1531 ± 54.74 ^a^	0.74 ± 0.19 ^a^	1.22 ± 0.23 ^a^	23.64 ± 7.74 ^a^	23.64 ± 7.74 ^a^
	Control ^3^	1475 ± 97.61 ^a^	0.78 ± 0.22 ^a^	1.24 ± 0.31 ^a^	23.61 ± 10.17 ^a^	23.61 ± 10.17 ^a^
	ZnSO_4_ ^3^	1436 ± 127.85 ^a^	0.64 ± 0.06 ^b^	1.02 ± 0.18 ^b^	18.39 ± 4.64 ^b^	18.39 ± 4.64 ^b^
	Zn-Gly ^3^	1324 ± 169.28 ^b^	0.54 ± 0.05 ^c^	0.85 ± 0.16 ^c^	14.12 ± 1.49 ^c^	14.12 ± 1.49 ^c^
*p-*value						
Temperature		<0.001	<0.001	<0.001	0.012	<0.001
Maternal zinc		0.003	<0.001	<0.001	0.001	<0.001
Temperature × maternal zinc		0.108	0.002	0.216	0.975	<0.001

ROS, reactive oxygen species. MDA, malondialdehyde. PC, protein carbonyl 8-OHdG, 8-hydroxydeoxyguanosine. HSP70, heat shock protein 70. Different letters (a–d) represent significant differences between the groups. ^1^ Each value represents the mean ± SE of 10 replicates (*n* = 10). ^2^ Each value represents the mean ± SE of 30 replicates (*n* = 30). ^3^ Each value represents the mean ± SE of 20 replicates (*n* = 20).

**Table 5 antioxidants-11-01699-t005:** Effects of maternal zinc on oxidative indicators and HSP70 in embryos under heat stress (mean ± S.E.).

Temperature	Maternal Zinc	T-SOD, U/mg port	CuZn-SOD, U/mg port	GSH-Px, U/mg port	CAT, U/mg port	T-AOC, U/mg port	MT, ng/g
37 °C ^1^	Control	93.26 ± 3.05 ^d^	63.67 ± 0.79 ^c^	73.52 ± 3.30 ^b^	8.02 ± 0.46 ^bc^	0.86 ± 0.07 ^c^	93.98 ± 0.37 ^c^
	ZnSO_4_	114.7 ± 4.18 ^ab^	67.29 ± 1.51 ^b^	76.32 ± 2.71 ^b^	8.71 ± 0.5 ^ab^	1.14 ± 0.05 ^ab^	96.84 ± 0.41 ^b^
	Zn-Gly	119.6 ± 4.91 ^a^	71.01 ± 0.80 ^a^	80.48 ± 0.16 ^a^	9.10 ± 0.57 ^a^	1.19 ± 0.07 ^a^	102.49 ± 0.87 ^a^
39.5 °C ^1^	Control	86.13 ± 1.74 ^e^	60.86 ± 0.17 ^d^	64.57 ± 0.44 ^d^	6.84 ± 0.45 ^d^	0.71 ± 0.05 ^d^	79.54 ± 2.19 ^e^
	ZnSO_4_	102.7 ± 1.08 ^c^	65.87 ± 1.84 ^b^	69.38 ± 1.45 ^c^	7.52 ± 0.37 ^cd^	0.84 ± 0.04 ^c^	90.75 ± 0.54 ^d^
	Zn-Gly	112.7 ± 3.73 ^b^	70.57 ± 0.46 ^a^	73.40 ± 1.29 ^b^	8.46 ± 0.33 ^ab^	1.05 ± 0.11 ^b^	93.60 ± 2.35 ^c^
37 °C ^2^		109.2 ± 12.66 ^a^	67.32 ± 3.32 ^a^	76.77 ± 3.71 ^a^	8.61 ± 0.65 ^a^	1.06 ± 0.16 ^a^	98.63 ± 3.91 ^a^
39.5 °C ^2^		100.7 ± 11.16 ^b^	65.77 ± 4.31 ^b^	69.12 ± 3.96 ^b^	7.60 ± 0.78 ^b^	0.87 ± 0.16 ^b^	87.96 ± 6.64 ^b^
	Control ^3^	89.69 ± 4.49 ^c^	62.27 ± 1.62 ^c^	69.04 ± 5.33 ^c^	7.43 ± 0.76 ^c^	0.78 ± 0.10 ^c^	86.76 ± 8.03 ^c^
	ZnSO_4_ ^3^	107.8 ± 6.90 ^b^	66.58 ± 1.69 ^b^	72.85 ± 4.27 ^b^	8.11 ± 0.76 ^b^	0.99 ± 0.17 ^b^	93.80 ± 3.36 ^b^
	Zn-Gly ^3^	116.2 ± 5.45 ^a^	70.79 ± 0.63 ^a^	76.94 ± 3.96 ^a^	8.78 ± 0.54 ^a^	1.12 ± 0.11 ^a^	99.16 ± 4.81 ^a^
*p-*value							
Temperature		<0.001	0.011	<0.001	0.001	<0.001	<0.001
Maternal zinc		<0.001	<0.001	<0.001	0.001	<0.001	<0.001
Temperature × maternal zinc		0.315	0.212	0.614	0.507	0.138	<0.001

T-SOD, total superoxide dismutase. CuZn-SOD, copper, and zinc superoxide dismutase. GSH-Px, glutathione peroxidase. CAT, Catalase. T-AOC, total antioxidant capacity. MT, metallothionein. Different letters (a–e) represent significant differences between the groups. ^1^ Each value represents the mean ± SE of 10 replicates (*n* = 10). ^2^ Each value represents the mean ± SE of 30 replicates (*n* = 30). ^3^ Each value represents the mean ± SE of 20 replicates (*n* = 20).

**Table 6 antioxidants-11-01699-t006:** Effects of maternal zinc on hepatocellular mitochondrial membrane potential, apoptosis Index and caspase3 activity in embryos under heat stress (mean ± S.E.).

Temperature	Maternal Zinc	MMP, mV	AI, %	Caspase3, U/mg Prot
37 °C ^1^	Control	12.06 ± 0.18	2.64 ± 0.21 ^d^	56.35 ± 1.80
	ZnSO_4_	13.13 ± 0.33	1.43 ± 0.15 ^e^	52.82 ± 1.28
	Zn-Gly	14.93 ± 0.84	0.24 ± 0.03 ^f^	49.25 ± 1.46
39.5 °C ^1^	Control	10.93 ± 0.60	7.38 ± 0.34 ^a^	68.17 ± 1.69
	ZnSO_4_	12.51 ± 0.31	5.60 ± 0.30 ^b^	63.59 ± 3.68
	Zn-Gly	13.68 ± 1.42	3.61 ± 0.24 ^c^	59.25 ± 2.81
37 °C ^2^		13.53 ± 1.38 ^a^	1.44 ± 1.01 ^b^	52.81 ± 3.18 ^b^
39.5 °C ^2^		12.37 ± 1.43 ^b^	5.53 ± 1.59 ^a^	63.67 ± 4.58 ^a^
	Control ^3^	11.49 ± 0.74 ^c^	5.01 ± 2.46 ^a^	62.26 ± 6.66 ^a^
	ZnSO_4_ ^3^	12.82 ± 0.45 ^b^	3.52 ± 2.15 ^b^	57.44 ± 6.20 ^b^
	Zn-Gly ^3^	14.39 ± 1.21 ^a^	1.92 ± 1.74 ^c^	54.25 ± 5.83 ^c^
*p-*value				
Temperature		0.013	<0.001	<0.001
Maternal zinc		<0.001	<0.001	<0.001
Temperature × maternal zinc		0.745	<0.001	0.781

MMP, Mitochondrial Membrane Potential. AI, Apoptosis Index. Different letters (a–f) represent significant differences between the groups. ^1^ Each value represents the mean ± SE of 10 replicates (*n* = 10). ^2^ Each value represents the mean ± SE of 30 replicates (*n* = 30). ^3^ Each value represents the mean ± SE of 20 replicates (*n* = 20).

**Table 7 antioxidants-11-01699-t007:** Effect of maternal zinc on CuZn-SOD, MT, Nrf2, UCP and Bax mRNA abundance expression in embryos liver under heat stress (mean ± S.E.).

Temperature	Maternal Zinc	CuZn-SOD	MT	Nrf2	UCP	Bax
37 °C ^1^	Control	0.64 ± 0.03 ^d^	0.87 ± 0.06 ^c^	0.93 ± 0.05	0.85 ± 0.13 ^d^	1.37 ± 0.05 ^c^
	ZnSO_4_	0.85 ± 0.05 ^b^	0.98 ± 0.03 ^c^	1.12 ± 0.10	0.99 ± 0.11 ^d^	0.99 ± 0.11 ^d^
	Zn-Gly	1.19 ± 0.03 ^a^	1.72 ± 0.18 ^a^	1.23 ± 0.05	1.03 ± 0.02 ^d^	0.95 ± 0.09 ^d^
39.5 °C ^1^	Control	0.60 ± 0.01 ^d^	0.82 ± 0.05 ^c^	1.12 ± 0.11	1.52 ± 0.15 ^c^	2.43 ± 0.27 ^a^
	ZnSO_4_	0.71 ± 0.00 ^c^	0.88 ± 0.10 ^c^	1.33 ± 0.02	2.08 ± 0.26 ^b^	2.08 ± 0.26 ^b^
	Zn-Gly	1.16 ± 0.02 ^a^	1.54 ± 0.09 ^b^	1.51 ± 0.15	2.43 ± 0.27 ^a^	1.52 ± 0.15 ^c^
37 °C ^2^		0.88 ± 0.25 ^a^	1.19 ± 0.41 ^a^	1.09 ± 0.14 ^b^	0.96 ± 0.12 ^b^	1.09 ± 0.21 ^b^
39.5 °C ^2^		0.86 ± 0.26 ^b^	1.08 ± 0.35 ^b^	1.32 ± 0.18 ^a^	2.01 ± 0.45 ^a^	2.01 ± 0.45 ^a^
	Control ^3^	0.63 ± 0.03 ^b^	0.85 ± 0.05 ^b^	1.03 ± 0.13 ^c^	1.18 ± 0.39 ^c^	1.90 ± 0.61 ^a^
	ZnSO_4_ ^3^	0.78 ± 0.08 ^b^	0.93 ± 0.09 ^b^	1.24 ± 0.12 ^b^	1.46 ± 0.61 ^b^	1.46 ± 0.61 ^b^
	Zn-Gly ^3^	1.18 ± 0.03 ^a^	1.63 ± 0.16 ^a^	1.37 ± 0.18 ^a^	1.73 ± 0.79 ^a^	1.23 ± 0.33 ^c^
*p*-value						
Temperature		<0.001	0.038	<0.001	<0.001	<0.001
Maternal zinc		<0.001	<0.001	<0.001	<0.001	<0.001
Temperature × maternal zinc		0.004	0.536	0.661	0.010	0.035

CuZn-SOD, copper and zinc superoxide dismutase. MT, metallothionein. Nrf2, nuclear factor erythroid 2-related factor 2. UCP, uncoupling protein. Bax; Bcl2-Associated X Protein. Different letters (a–d) represent significant differences between the groups. ^1^ Each value represents the mean ± SE of 10 replicates (*n* = 10). ^2^ Each value represents the mean ± SE of 30 replicates (*n* = 30). ^3^ Each value represents the mean ± SE of 20 replicates (*n* = 20).

## Data Availability

Data is contained within the article.

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
