# Peer review of "Mechanisms Underlying the Protective Effect of Maternal Zinc (ZnSO4 or Zn-Gly) against Heat Stress-Induced Oxidative Stress in Chicken Embryo"

_antioxidants, 2022, doi:10.3390/antiox11091699_

Round 1
Reviewer 1 Report
This is a nice written paper devoted to an important research issue.
There are some limitation of this study. First, there is no data on Zn concentration in the diet and in egg and embryos. It is well known that Zn is not well transferred to the egg from the maternal diet (see
Mayer, A. N., Vieira, S. L., Berwanger, E., Angel, C. R., Kindlein, L., França, I., & Noetzold, T. L. (2019). Zinc requirements of broiler breeder hens. Poultry science, 98(3), 1288–1301) Without data on Zn level in tissues it is difficult to properly discuss these results. Was the control diet Zn deficient? Was the used dose of Zn supplementation sufficient? and many other questions are without the answer.
The second limitation is that stress model used (increased temperature) did not affect embryonic mortality. Therefore in this case decreased mortality by Zn supplementation is highly surprised.
It is necessary to present the number of samples analysed in each table (n=?). For example data on SOD (Table 5: 70.57±0.46a 73.40±1.29b) have very low biological variability (M+/-m) e.g. 0.46 (from my experience this should be 5-10fold higher), therefore, a significance of this difference (70.6 vs 73.4) is questionable.
Reviewer 2 Report
I am glad to review this kind of paper, it is interestion to focus on the effect of maternal nutrinent level on the offspring performance. I have some questions about your research. The major one is that your result of Embryo Mortality (table 3) indicated that short-trem of high incubation temperature would not influenced the embryo mortality, even though the oxidation indicators showed the oxdiation status changed. Thus I wonder whether it can be said that zinc can fix the oxidative damage in this study, because mortility did not change among the temperature groups. In other words, the connection between incubation temperature and oxidative damage is too weak to support your discussion and conclusion. And there are some small questions:
Line 86, "each group containing 5 replicates and 20 replicates each". The two "repilcates" here could make misleadings. The following words present "replicate" again, and it would take time to understand which one it refered to.
Line 93, "all birds were artificially inseminated". Were the semen from one male brolier breeder?How to control the influence of male?
Line 213. Check you data in Table 4, there should not be a decimal point in the first line of ROS data.
Line 288-289. The figure 3 was a great conclusion of your work, I suggest to put it in the later part of your discussion and the discussion could follow the logistc way.
Line 312, "Who". It should be the lowercase.
Line 313, "d10 to d18". Using a consistent way to describe the day of age.
Reviewer 3 Report
Dear authors,
For starters I want to congratulate you for the article.
There are some negative aspects, which I believe need to be clarified and corrected in order to publish this manuscript:
- Lines 83-97: In the Experimental Design, Birds, and Diets chapter, please specify the source of organic zinc (Zn-Gly).
- Enzyme activity and other parameters are expressed in U/mg protein when tissue protein extracts are analyzed. It is unclear if and how the dosage of total proteins from the tissue extracts was carried out.
- Table 4 contains HSP70 expressed as ng/g. What does g mean? Tissue?
- Lines 135 – 150, Mitochondrial Membrane Potential (MMP) - please rephrase, the protocol is not clear.
- Lines 151-161, Assessment of Hepatocyte Apoptosis - please rephrase, the protocol is not clear.
- Lines 229: T-AOC represents the total antioxidant capacity (not total antioxidation capability)
- Line 248 – Caspase3 activity is specified as results. How was it determined? There is no mention of this enzyme in the materials and methods chapter.
- Figure 1. nd figure 2 – they are of very poor quality; the morphological aspects cannot be clearly distinguished.
- Lines 297-305: Several question marks and inconsistencies appear. First of all, T-AOC represents the total antioxidant capacity that is not only given by antioxidant enzymes, it also includes other non-enzymatic antioxidant molecules that are found in the liver. Also, in this study, only the expression of the specific gene for CuZn-SOD was investigated, the other specific genes for CAT and GSH-Px were not studied, so we cannot say with certainty that the administration of Zn (in inorganic or organic form) induces changes in their expression.
- Lines 321-323: In oxidative stress induced in this case by temperatures, the production of ROS is intensified. An accumulation of them can increase the level of oxidative degradation of the protein structure of the enzymes, which can lead to a decrease in their activity (through degradation at the level of the active site). „The changed activity of these antioxidant enzymes is thought to be a protective response to oxidative stress” It is not a correct statement in my opinion. I consider that the increase in activity can be a protective response, not its decrease.
- Line 327-328: Because of the levels of MDA, PC, and 8-OHdG can be used as markers to assess the oxidative damage degree of protein, DNA and lipid in animals. Please rephrase, it is not clear and correct.
Round 2
Reviewer 1 Report
The authors did not address my concern about Zn level in egg and embryonic tissues depending on Zn sources. The reference to previous work is not a solution. If the authors do not have such data, they should present clear evidence from other publications about Zn transfer from the feed to the egg and further to the developing embryo. This is a milestone question, without proper addressing it it is difficult to judge research value of the paper. I would suggest to add a references to papers of other authors showing limited transfer of Zn to the egg, demonstrate dose response, mention possible mechanisms of transfer and explain HOW Zn level is increasing in embryonic liver, if its concentration in the egg did not respond well to dietary Zn. If the authors could prove the point that without analysing Zn content in the egg and liver they can ensure that Zn concentration in the embryonic liver significantly and substantially increased due to dietary Zn supplementation, the paper can be assessed further.
Reviewer 2 Report
I am satisfied with your response.
Author Response
Thank you for your approval.
Reviewer 3 Report
I read the manuscript and noticed that the authors made the requested changes. The quality of the article resubmitted has increased.
Author Response
Thank you for your approval.
Round 3
Reviewer 1 Report
The paper is accepted in the present form